# Cancer-Associated Fibroblasts Modulate Transcriptional Signatures Involved in Proliferation, Differentiation and Metastasis in Head and Neck Squamous Cell Carcinoma

**DOI:** 10.3390/cancers13133361

**Published:** 2021-07-04

**Authors:** Emilia Wiechec, Mustafa Magan, Natasa Matic, Anna Ansell-Schultz, Matti Kankainen, Outi Monni, Ann-Charlotte Johansson, Karin Roberg

**Affiliations:** 1Division of Cell Biology, Department of Biomedical and Clinical Sciences, Linköping University, 58185 Linköping, Sweden; emilia.wiechec@liu.se (E.W.); mostafa.magan@gmail.com (M.M.); natasa.matic@regionostergotland.se (N.M.); anna.ansell.schultz@dynamiccode.se (A.A.-S.); ann-charlotte.johansson@liu.se (A.-C.J.); 2Department of Otorhinolaryngology in Linköping, Anesthetics, Operations and Specialty Surgery Center, Region Östergötland, 58185 Linköping, Sweden; 3Translational Immunology Research Program and Department of Clinical Chemistry, University of Helsinki, 00290 Helsinki, Finland; matti.kankainen@helsinki.fi; 4Department of Medical and Clinical Genetics, University of Helsinki, Helsinki University Hospital, 00029 Helsinki, Finland; 5iCAN Digital Precision Cancer Medicine Flagship, University of Helsinki, 00014 Helsinki, Finland; outi.monni@helsinki.fi; 6Applied Tumor Genomics Research Program and Department of Oncology, Faculty of Medicine, University of Helsinki, 00014 Helsinki, Finland

**Keywords:** HNSCC, cancer-associated fibroblasts, cDNA microarray, MMP9, FMOD, overall survival

## Abstract

**Simple Summary:**

Cancer-associated fibroblasts (CAFs) are the major cellular component of the tumor microenvironment and have been shown to stimulate tumor growth, epithelial-to-mesenchymal transition (EMT), invasion, and radio-resistance. Radio-resistance leading to disease relapse is one of the major challenges in long-term survival and outcome in head and neck squamous cell carcinoma (HNSCC). Therefore, it is essential to search for predictive markers and new targets for treatment using clinically relevant in vitro tumor models. We show that CAFs alter the expression of HNSCC tumor cell genes, many of which are associated with proliferation, differentiation, and metastasis. Moreover, the expression pattern of selected CAF-regulated genes differed between HNSCC tumor tissue and the adjacent non-tumoral tissue. Two CAF-regulated genes, MMP9 and FMOD, were found to be associated with overall survival (OS) in patients treated with radiotherapy.

**Abstract:**

Cancer-associated fibroblasts (CAFs) are known to increase tumor growth and to stimulate invasion and metastasis. Increasing evidence suggests that CAFs mediate response to various treatments. HNSCC cell lines were co-cultured with their patient-matched CAFs in 2D and 3D in vitro models, and the tumor cell gene expression profiles were investigated by cDNA microarray and qRT-PCR. The mRNA expression of eight candidate genes was examined in tumor biopsies from 32 HNSCC patients and in five biopsies from normal oral tissue. Differences in overall survival (OS) were tested with Kaplan–Meier long-rank analysis. Thirteen protein coding genes were found to be differentially expressed in tumor cells co-cultured with CAFs in 2D and 81 in 3D when compared to tumor cells cultured without CAFs. Six of these genes were upregulated both in 2D and 3D (*POSTN, GREM1, BGN, COL1A2, COL6A3*, and *COL1A1*). Moreover, two genes upregulated in 3D, MMP9 and FMOD, were significantly associated with the OS. In conclusion, we demonstrated in vitro that CAF-derived signals alter the tumor cell expression of multiple genes, several of which are associated with differentiation, epithelial-to-mesenchymal transition (EMT) phenotype, and metastasis. Moreover, six of the most highly upregulated genes were found to be overexpressed in tumor tissue compared to normal tissue.

## 1. Introduction

Cancer of the head and neck is the seventh most common form of cancer worldwide. About 90% of cases are head and neck squamous cell carcinomas (HNSCC), which have their origin in the mucosa of the oral cavity, pharynx, larynx, and nasal cavity [1]. The HNSCC cancer treatment is primarily based on a combination of surgery and radiotherapy, but during the last decades, supplementary treatments such as cisplatin-based chemotherapy and molecular targeted therapy have been introduced. Despite therapy improvements, the five-year overall survival (OS) in patients with regional metastasis HNSCC has not changed much and is still less than 50% [2]. Therefore, it is a great need for new therapeutic strategies and biomarkers that are predictive of the treatment response.

Rather than being understood as malignant cells growing in isolation, cancer has over the past decades been recognized as a complex tissue where many different cell types and extracellular matrix interact in a multipart ecosystem. This results in the creation of a distinct microenvironment within the tumor [3]. One of the most crucial elements of the microenvironment is the cancer-associated fibroblasts (CAFs). These cells modulate the tumor’s fate by increasing tumor growth [4], epithelial-to-mesenchymal transition (EMT) [5], the invasive potential [6], and metastasis [7] by secretion of soluble factors or by modification of extracellular matrix components [8]. In addition, CAFs have also been suggested to modulate the drug sensitivity of cancer cells [9,10]. High density of CAF within the tumor microenvironment is associated with tumor progression and vascular and perineural invasion that correlate with high rate of local recurrence [11,12]. The role of CAFs is not fully understood in HNSCC, but different subgroups of CAFs that possess different properties such as stimulation of Matrigel invasion and tumorigenicity have been identified [13]. The tumor genetic background may have impact on CAF phenotype, and genetically unstable tumor cells have been shown to induce senescence in CAFs [14]. Furthermore, CAFs from different patients have been shown to have a different “sensitizing ratio” for cisplatin response in HNSCC cells in vitro [15].

We have shown earlier that CAFs increase resistance to cetuximab treatment (a monoclonal antibody against EGFR; Erbitux^®^, Merck KGaA) and increase the MMP1 expression of HNSCC cells during co-culturing compared with tumor cells grown alone in a 2D model [9]. Recently, we showed that CAFs co-cultured with HNSCC cells in spheroids increase the tumor cell proliferation and have positive impact on the cetuximab and cisplatin treatment response [16]. A better understanding of the HNSCC microenvironment would promote the development of more effective treatments. Hence, it is crucial to study how CAFs influence the gene expression, proliferation, and migration of HNSCC cells.

This study was undertaken to investigate the influence of CAFs on the gene expression profile of tumor cells in 2D and 3D co-cultures, and furthermore to assess the potential impact of CAF-regulated genes on radiotherapy response and OS of the patients diagnosed with HNSCC.

## 2. Materials and Methods

### 2.1. Cells and Culture Conditions

Eight HNSCC cell lines—LK0824, LK0858, LK0917, LK0902, LK0923, LK0942, LK0949, and LK1108 (Appendix A)—which were previously established [17] from tissue specimens from patients diagnosed with HNSCC, were grown in Keratinocyte-SFM supplemented with antibiotics (50 U/mL of penicillin and 50 µg/mL of streptomycin) and 10% fetal bovine serum (FBS; all from Gibco, Invitrogen Corporation, Paisly, UK). The cells were given fresh culture media twice per week and were subcultured at confluence after detaching the cells with 0.25% trypsin +0.02% EDTA at a weekly split ratio of approximately 1:2 or 1:3. Cultures from passages 10 to 25 were used in all experiments.

Eight CAFs were isolated as previously described [9] and originated from the same eight HNSCC patients as the tumor cell lines listed above. CAFs cultures were grown in Keratinocyte-SFM supplemented with antibiotics (50 U/mL of penicillin and 50 µg/mL of streptomycin) and 10% FBS (all from Gibco) and were given fresh media twice per week. For all experiments, CAFs were used at passages 3–6.

The cells were screened periodically for mycoplasma contamination using the MycoAlert Mycoplasma Detection Kit (Lonza, Walkersville, MD, USA).

### 2.2. Co-Culture of Tumor Cells and CAFs

For 2D culturing, tumor cells and CAFs were co-cultured in cell culture flasks for 7 days. Tumor cell and CAF numbers were adjusted so that an approximate ratio of 3:1 was achieved by the end of co-culture.

As previously described, to achieve spheroids of tumor cells or co-cultures of tumor cells and CAFs, we seeded 15,000 tumor cells or 10,000 tumor cells with 5000 CAFs into each well of ultra-low attachment (ULA) 96-well round-bottomed plates (Corning, Amsterdam, The Netherlands) and were thereafter cultured in standard culture conditions [16].

### 2.3. Magnetic Activated Cell Sorting (MACS)

Tumor cells grown both in 2D and 3D cultures were separated from CAFs by magnetic activated cell sorting (MACS) [16]. RNA was isolated from tumor cells and further processed for microarray and qPCR analysis (as described below). Tumor cells cultured without CAFs served as controls.

Cells growing in 2D were incubated with trypsin for 5 min and thereafter flushed in culture media and then subjected to separation. Spheroids were sampled in a test tube, centrifuged (1000 rpm, 5 min), washed in PBS, centrifuged, and then incubated with 500 µL trypsin for 5 to 20 min at 37 °C. Thereafter, 1500 µL culture media was added, and the cells were flushed 30 times with a micropipette and then subjected to separation.

The tumor cells or the tumor cells/CAFs cell suspensions were magnetically labelled with Anti-fibroblast Micro Beads (Miltenyi Biotech Norden, Lund, Sweden) by a 15-min incubation at 4 °C. The cells were washed, suspended in PBS, and passed through a 50 µm filter (Becton Dickinson, Franklin Lakes, NJ, USA). The cell suspensions were then loaded onto MACS^®^ Column Beads (Miltenyi Biotech). The tumor cells were collected in test tubes placed under the column, and the CAFs were collected in the column due to the magnetic field. This procedure was repeated three times with each suspension to increase the purity of each population. To check the purity of the tumor cell suspensions, we cultured cells for 48 h, and thereafter 500 cells with epithelial or fibroblast morphology were counted in a phase contrast light microscope. Less than 1% of the cells were found with a fibroblast morphology in all samples.

### 2.4. Microarray Analysis

HNSCC cell lines and their patient-matched CAFs were selected for the microarray analysis in order to compare gene expression differences between tumor cells co-cultured with patient-matched CAFs and tumor cells cultured alone. Briefly, 2D and 3D microarray design included in total 5 and 3 biological replicates, respectively. In both cases, multiple independently derived HNSCC cell lines were treated as biological replicates. Each experimental condition (−CAFs/+CAFs) included a single hybridization of the same sample where condition was represented by multiple cell lines (*n* = 5/condition, 2D microarray design; *n* = 3/condition, 3D microarray design).Total RNA was prepared using the RNeasy Mini Kit (Qiagen, Valencia, CA, USA), and the quality was verified using the Agilent 2100 Bioanalyzer (Agilent Technologies, Santa Clara, CA, USA)and quantified using a Nanodrop ND-100 Spectrophotometer (Thermo Fisher Scientific, Waltham, MA, USA).

For the 2D microarray analysis, five HNSCC cell lines (LK0824, LK0858, LK0923, LK0942, LK0949) and their patient matched CAFs were selected. The labeling and hybridization were done according to the GeneChip^®^ WT Terminal Labeling and Hybridization User Manual for use with the Ambion^®^ WT Expression Kit (Affymetrix, Santa Clara, CA, USA). The starting amount of total RNA was 100 ng. A total of 15 µg of cRNA was used for single-stranded cDNA synthesis (sscDNA). A total of 5.5 µg of sscDNA was fragmented and hybridized on a Human Exon 1.0 ST Array.

For the 3D microarray analysis, three HNSCC cell lines (LK0902, LK0917, and LK1108) and their patient matched CAFs were selected, and 150 ng of total RNA was used to process the Affymetrix GeneChip^®^ Human Transcriptome 2.0 Arrays using the GeneChip^®^ WT Plus Reagent Kit according to the manufacturer’s instructions (Thermo Fisher Scientific, Waltham, MA, USA). Hybridized arrays were scanned with an Affymetrix GeneChip^®^ 3000 fluorescent scanner.

Microarray data were analyzed with Transcription Analysis Console (TAC) v. 4.0.2.15 (Thermo Fisher Scientific) with default settings. Briefly, the SST-RMA was used for normalization of the 3D microarray data and RMA for normalization of 2D microarray data. Differential expression analysis was carried out with Limma package in TAC with a paired design. Probesets with the detected above background (DABG) value < 0.05 in less than half of samples were not considered. Gene fold change <−2 or >2 were considered differentially expressed. Heatmaps were generated with a freely available online application Heatmapper [18]. Description of samples subjected to microarray analysis is provided in Appendix A. The data have been deposited at Gene Expression Omnibus (http://www.ncbi.nlm.nih.gov/geo/ (accessed on 15 June 2021)) database under the accession numbers GSE178153 and GSE178154.

### 2.5. qRT-PCR

The qRT-PCR analysis was performed on a 7500 Fast Real-Time PCR system (Applied Biosystems, Waltham, MA, USA). Total RNA was extracted from cells using the RNeasy Mini Kit (Qiagen, Hilden, Germany), cDNA was synthesized using the High-Capacity RNA-to-cDNA Kit (Applied Biosystems), and TaqMan FAM/MGB probes (Applied Biosystems) were used for PCR reactions. Amplification of two housekeeping genes, glyceraldehyde 3-phosphate dehydrogenase (GAPDH) and ß-actin, was used as an internal standard. The data were calculated according to the comparative Ct method to present the data as fold differences in the expression levels relative to the control sample [19].

### 2.6. Immunohistochemistry (IHC)

Spheroids were fixed in 4% paraformaldehyde (Santa Cruz Biotechnology, Dallas, TX, USA) overnight at 4 °C followed by rinsing with PBS, stained with 0.1% toluidine blue D (Merck, Kenilworth, NJ, USA) and centrifuged in 2% agarose. Agarose blocks containing spheroids were dehydrated and thereafter embedded in paraffin wax (Merck, Kenilworth, NJ, USA). Sections of 5 μm were collected on Super Frost Plus slides (Thermo Fisher Scientific, Fremont, CA, USA), dried overnight, and then incubated at 58 °C for 1 h. Sections were deparaffinized in Histolab-Clear (HistoLab, Gothenburg, Sweden), re-hydrated in graded alcohols, underwent antigen retrieval (PT 200, DAKO, Glostrup, Denmark A/S), and were then blocked for endogenous peroxidase activity in 3% peroxidase (Sigma-Aldrich, St. Louis, MO, USA). After incubation in blocking buffer (0.1% BSA-5% FBS in TBS-Triton), sections were incubated overnight at 4 °C with primary antibodies: MMP1 (1:50; Abcam, Cambridge, UK), MMP9 (1:500, Abcam), gremlin (1:100, Abcam), periostin (1:50), Abcam), and fibromodulin (1:50; Invitrogen, Carlsbad, CA, USA). Thereafter, sections were washed in TBS-triton buffer and incubated with a goat anti-rabbit or a goat-anti-mouse HRP-antibody (1:500; IgG, EMD Millipore Corporation, Temecula, CA, USA) for 60 min at room temperature. Following the washing steps, slides were developed for 6 min using the ImmPACT NovaRED peroxidase substrate kit at room temperature (Vector Laboratories, Burlingame, CA, USA). Thereafter, sections were counterstained in Mayer’s hematoxylin (HistoLab) for 4 min, dehydrated, cleared in Histolab-Clear, and cover slipped with Pertex (HistoLab). As negative controls sections were stained as above described but without the primary antibody. All negative controls were revealed unstained. Images were acquired with a light microscope (Olympus BX51, Shinjuku City, Tokyo, Japan) with a 20× objective.

### 2.7. Tumor Material and Patient Characteristics

Samples from patients with HNSCC were obtained from an established tumor collection (No 416, The National Board of Health and Welfare in Sweden) at the Department of Otorhinolaryngology, Head and Neck Surgery, at the University Hospital of Linköping, Sweden (approved by the Ethical Committee of Linköping). The collection consists of approximately 350 tumor biopsies, freshly obtained at the time of surgery after obtaining due consent from the patients. Tumor and CAFs cell lines were established from one part of pre-treatment biopsies [17]: one part of the biopsy was immediately frozen in liquid nitrogen and stored in −70 °C, and the third part was formalin-fixed and paraffin-embedded. For all samples, clinical and follow-up data of the patients were available from the Department of Otorhinolaryngology.

All patients included received radiotherapy as follows: 13 (40.6%) received radiotherapy alone, 15 (46.9%) received preoperative radiotherapy, 3 (9.4%) received postoperative radiotherapy, and 1 (3.1%) received chemotherapy in addition to radiotherapy. Patients were retrospectively divided into two groups according to treatment response. A patient was considered as a responder if the tumor size was reduced during radiotherapy and no sign of recurrent disease was noted within one year following radiation. In the non-responder group, tumors either grew during on-going radiation or patients were diagnosed with relapse of disease within 6 months of radiotherapy. Sixteen individuals were identified as non-responder, whereafter sixteen responders were selected and matched according to tumor localization and histological grade as closely as possible (Appendix A). The distribution of patients according to TNM classification (according to the general rules of head and neck cancer), histological grade, sex, and age has earlier been published [20]. Non-tumoral biopsies were collected from the sites as far as possible from the tumor margin.

### 2.8. Statistical Analysis

Statistical analyses were performed with Prism 7.0 (GraphPad Software, Inc., La Jolla, CA, USA) and SPSS IBM version 26.0 (IBM Corporation, Armonk, NY, USA). All values obtained were represented as mean ± SD of at least three independent experiments. For the verification of microarray in HNSCC cell lines, log_2_ mRNA expression levels were analyzed using unpaired two-tailed Student’s *t*-test. One-way ANOVA followed by post-hoc Tukey’s multiple correction test was applied to analyze the patient data. The mRNA expression results of selected genes were analyzed for correlation with OS using the Kaplan–Meier survival analysis with log-rank test. The relationship between mRNA expression in tumor tissue and treatment response was assessed by binary logistic regression. The high and low mRNA levels were assessed on the basis of the median mRNA expression. All values obtained were represented as mean ± SD of at least three independent experiments.

## 3. Results

### 3.1. Microarray Analysis in 2D Cultures

Five HNSCC cell lines (LK0824, LK0858, LK0923, LK0942, and LK0949; Appendix A) were co-cultured in 2D with their patient matched CAFs for seven days. CAFs were distributed around tumor cell colonies in a similar pattern as observed in patient derived tumor samples (Figure 1A). Tumor cells cultured without CAFs served as controls (Figure 1B). Tumor cells were separated from CAFs by MACS, and the purity of the tumor cell cultures were verified by microscopy. The gene expression of the five HNSCC cell lines co-cultured with or without their patient matched CAFs was investigated by microarray analysis.

The microarray data analysis revealed eight genes that were upregulated while 38 genes were downregulated (Table 1 and Appendix A). The majority of these expression differences were deduced on the basis of the fold-change alone and had a relatively large biological variability due to the treating of independently derived cell line as biological replicates in the experiment.

### 3.2. Microarray Analysis in 3D Cultures

Recently, we have developed a unique tumor experimental in vitro model that consists of low-passage HNSCC cells (LK0902, LK0917, and LK1108) and CAFs from the same tumor growing in spheroids. In this in vitro model, CAFs affect proliferation; EGFR expression; and, to some extent, the EMT and cancer stem cell phenotype of HNSCC tumor cells [16].

Here, we used this biologically relevant in vitro model and investigated which tumor genes are affected by CAFs. Three HNSCC cell lines (LK0902, LK0917, and LK1108) were co-cultured in spheroids with their patient-matched CAFs for five days. Thereafter, tumor cells were separated from CAFs by MACS, and the purity of the tumor cell cultures were verified by microscopy. The gene expression profiles in the three HNSCC cell lines co-cultured with or without their patient matched CAFs was investigated by microarray analysis.

In tumor cells co-cultured with CAFs, 164 genes were upregulated while 169 genes were downregulated (Table 1 and Appendix A) compared to tumor cells grown without CAFs. Similar to the 2D experiment, expression differences were associated with a relatively large cell line to cell line variability and calling relied on the fold-change cut-offs. Interestingly, six genes—*POSTN*, *GREM1*, *BGN*, *COL1A2*, *COL6A3*, and *COL1A1*—that were found to be upregulated in 2D cultures were also upregulated in spheroids. In contrast, no shared downregulated genes were found between 2D and 3D cultures.

### 3.3. Verification of Microarray Results Using qRT-PCR

Verification of microarray results was performed using qRT-PCR. Three of the differentially expressed protein coding genes (*COL1A2, GREM1*, and *POSTN*) revealed by microarray analyses in 2D cultures were analyzed. The analysis verified a change in expression of *COL1A2, GREM1*, and *POSTN* upon co-culture with CAFs (Figure 2A). The expression of *GREM1* and *COL1A2* could not be detected in LK0824, LK0858, or LK0923 tumor monocultures by qRT-PCR; thus, the Ct-values were set to 40 in order to enable fold-change calculations using the comparative Ct method.

Moreover, 10 of the differentially expressed protein coding genes (*MMP1, MMP9, COL1A2, GREM1, POSTN, FMOD, THSB, DCN, VCAN, CTSK*, and *IVL*) revealed by microarray analysis of tumor cells cultured in spheroids were selected for verification by qRT-PCR. The analysis verified a significant upregulation of *MMP9, MMP1, COL1A2, GREM1, POSTN, FMOD, DCN, THSB1*, and *VCAN* mRNA expression in all cell lines upon co-culture with CAFs. Furthermore, *CTSK* was upregulated in two of the cell lines, and IVL was found highly downregulated in all cell lines (Figure 2B–D).

### 3.4. Verification of Microarray Results on Protein Level

To further verify the microarray results, the protein expression of MMP9, MMP1, periostin, gremlin 1, and fibromodulin were investigated in spheroids with immunohistochemistry (IHC). MMP1 was found to be upregulated in all three cell lines when co-cultured with CAFs compared to tumor cells cultured alone in spheroids (Figure 3 and Appendix A). MMP9 was upregulated in LK0902 and LK0917 tumor cells/CAFs spheroids, but in LK1108, only a few cells were positively stained for MMP9 (Figure 3 and Appendix A). Periostin and gremlin showed a weak to moderate staining in tumor cell/CAF spheroids compared to tumor cell spheroids (Figure 4, Appendix A). Moreover, fibromodulin showed a moderate staining in LK0902 and LK1108 in tumor cell/CAF spheroids compared to tumor cell spheroids (Figure 4 and Appendix A). Despite the moderate staining of fibromodulin in LK0917 tumor spheroids, a relatively stronger staining was observed in tumor cell/CAF spheroids (Figure 4 and Appendix A).

### 3.5. Expression of CAF-Regulated Genes in Tumor Tissue and Their Impact on Radiotherapy Response

To determine the significance of CAF-regulated genes in HNSCC, we analyzed the mRNA expression of most dysregulated genes including *MMP1, MMP9, POSTN, GREM1, FMOD, COL1A2, GREM1, IVL*, and *α-SMA* (a CAF marker) in two groups of HNSCC patients (responders or non-responders to radiotherapy) and in five biopsies from adjacent normal oral tissue.

Our results show a significantly higher mRNA expression of *MMP1, MMP9, POSTN, GREM1, FMOD*, and *COL1A2* in tumor biopsies compared to normal tissue (Figure 5), but the significant difference between responder and non-responder groups was only found for the expression of *FMOD* mRNA. The expression of *α-SMA* was higher in the HNSCC biopsies than in adjacent normal oral tissue reaching the statistical significance in the responder group.

We next tested whether mRNA levels of analyzed genes correlate with OS of HNSCC patients. Patients with high expression of *MMP9* mRNA had significantly longer OS compared to the low expression group (*p* = 0.011; Figure 6A). Like MMP9, patients with high expression of *FMOD* mRNA had significantly longer OS compared with those who had low expression of *FMOD* mRNA (*p* = 0.026; Figure 6B). Moreover, the combination of high expression of both *MMP9* and *FMOD* mRNA was associated with significantly longer OS compared to low and/or high expression of either *MMP9* or *FMOD* in the analyzed cohort (*p* = 0.017; Figure 6C). There was a trend towards a longer OS for patients expressing a high level of *IVL* mRNA (*p* = 0.33; Figure 6D).

## 4. Discussion

For many years, cancer research has focused mainly on the cancer cell and its molecular changes; however, the stromal microenvironment is now generally accepted to modulate the tumor’s fate by increasing tumor growth [9,10,11], the invasive potential [14,15], and metastasis [12,13,14,15,16] by secretion of soluble factors or by modification of extracellular matrix components [17]. The molecular mechanisms behind these CAF-induced changes in the tumor cell phenotype are complex and still not fully understood.

In this study, we investigated the influence of CAFs on the gene expression of five HNSCC cell lines growing in 2D and in three HNSCC cell lines growing in 3D. One of the strengths of our experimental plan lies in the use of several patient matched HNSCC cell lines and CAFs. The two cell types were grown in co-cultures allowing cell–cell contact both in 2D and 3D, which mimics the in vivo situation but to a higher extent in 3D. In this way, the crosstalk between tumor cells and CAFs can be closely monitored without the influence of other cell types present within the tumor microenvironment. By whole-genome microarray analysis, we were able to identify multiple genes that were differentially expressed in across multiple HNSCC cell lines co-cultured with CAFs as compared to monocultures.

Six genes (*POSTN, GREM1, BGN, COL1A2, COL6A3*, and *COL1A1*) were upregulated in both the 2D and 3D model. Recently it was published that 10 HUB genes (*SPP1, POSTN, COL1A2, FN1, IGFBP3, APP, MMP3, MMP13, CXCL8*, and *CXCL12)* were differentially expressed in HNSCC [21]. In one of our previous studies, *FN1* was found to be upregulated in two of the investigated cell lines [16]. In the present study, *POSTN* and *COL1A2* were upregulated in all cell lines after coculture with CAFs. It has been reported that *COL1A1* and *COL1A2* are both overexpressed in gastric cancer, and a high *COL1A1* and *COL1A2* mRNA expression was suggested to be a prognostic factor predicting OS [22]. Furthermore, *COL1A2* has been suggested to be a regulator of pancreatic cancer [23]. In the present study, we found six members of the collagen family to be upregulated tumor cells co-cultured with CAFs. Two of these were from the type I collagen group, which are mostly found in connective tissue and embryonic tissue [24], and are key structural components of the extracellular matrix.

When investigating the mRNA expression of *COL1A2, POSTN, GREM1, MMP1*, and *MMP9* in patient material, we found all five genes to be upregulated in HNSCC tumor tissue compared to normal oral tissue. Periostin (*POSTN*) is a protein ligand to transmembrane receptors called integrins. Integrins are important in adhesion between cells and between cells and extracellular matrix in tissue. Upregulation of periostin has been observed in a wide variety of cancers including colon, gastric, and esophageal cancer [25]. Gremlin-1 (*GREM1*) is a member of the cystine knot superfamily and is a bone morphogenetic protein (*BMP*) antagonist. BMPs play a critical role in embryogenesis, development, tissue differentiation, and regulation of the immune system [26]. *GREM1* is thought to be involved in EMT [27] and has been shown to be overexpressed in human tumors, including carcinomas of the colon and lung [28,29]. In breast cancer, *GREM1* was shown to be an oncogenic protein by promoting tumor cell proliferation, migration, and invasion, which is associated with worsening of survival. Moreover, knock-down of *GREM1* was found to result in decreased proliferation of breast cancer cells [30].

Matrix metalloproteinases (MMPs), which are known to degrade the extracellular matrix, have been suggested as strong mediators of cancer invasion and metastasis. MMPs are thought to regulate cancer cell growth, apoptosis, and angiogenesis by releasing bioactive fragments, and increased expression of MMPs has been shown to result in poor clinical outcome in several cancer types [31,32,33]. In a previous study, we demonstrated that *MMP1* was upregulated in HNSCC cells when cocultured with CAFs in a 2D model and that MMPs were involved in cetuximab resistance [9]. In spheroids, *MMP1* and *MMP9* showed the highest level of overexpression on the mRNA level and were found to be upregulated also at protein level using IHC. MMP1 was earlier shown to be upregulated in HNSCC tissue [34] and was positively associated with lymph node metastasis (LNM) [35,36]. Moreover, high expression of MMP9 has been reported to correlate to LNM, angiogenesis, and worse survival in HNSCC [37,38]. *MMP1* and *MMP9* were not upregulated in tumor cells co-cultured with CAFs in 2D, but these MMPs were upregulated in tumor tissue from HNSCC patients compared to normal oral tissue. These results indicate that that our 3D in vitro model is more similar to tumor tissue than our 2D model and that the upregulation of these MMPs is cell adhesion dependent. Surprisingly, in this study, patients with a high mRNA expression of MMP9 had a significant better survival compared with patients with a low expression (Figure 6A). The role of MMPs in cancers is more complex than previously thought. More than 50 MMP inhibitors have been investigated in various cancers, but all have failed [39], and in a study consisting of 37 HNSCC patients treated with cisplatin and cetuximab, a high mRNA expression of *MMP9* was associated with objective treatment response [40]. However, the impact of MMP9 as a predictive biomarker for radiotherapy response need to be further evaluated.

The small leucine-rich proteoglycan decorin (*DCN*) is an important component of the cellular microenvironment and has been shown to inhibit the growth of a wide variety of tumor cells, probably due to an interaction of the decorin core protein with EGFR [41]. Another small leucine-rich proteoglycan, fibromodulin (*FMOD*), was also upregulated at transcript level in tumor cells after coculturing with CAFs. Fibromodulin has been shown to affect biological processes such as angiogenesis, apoptosis, and migration, and exerts its effects via targeting certain signaling pathways such as TGF-β [42]. *FMOD* has been recognized as a novel tumor-associated antigen in lymphoma, leiomyoma, and leukemia [43], and *FMOD* has been detected in various clinical malignancies, such as breast, prostate, and lung carcinomas [44,45,46]. In this study, patients in the responder group had a significant higher expression of *FMOD* (Figure 5), and patients with a higher mRNA expression of *FMOD* had a significantly better survival compared with patients with a low expression (Figure 6B). The role of fibromodulin and decorin for radiotherapy response in HNSCC is still unknown, but in a breast cancer cell line, 4T1, overexpression of *FMOD* and *DCN* was associated with downregulation of NF-κB and TGF-β1 [47]. Activation of nuclear factor-κB (NF-κB) has been shown to inhibit apoptosis and to stimulate proliferation [48]. Furthermore, NF-κB signaling has also been shown to be essential for epithelial–mesenchymal transition and metastasis [49]. One theory for the higher survival rate observed in patients with a high *FMOD* expression could relate to an effect of *FMOD* on NF-κB and TGF-β1, but this has to be further investigated.

Several genes associated with cornified squamous epithelia and terminal differentiation were found to be downregulated in tumor cells cocultured with CAFs. One of them, sciellin, a precursor of the cornified envelope, has been found to mediate mesenchymal-to-epithelial transition and knockdown of sciellin increase invasion and migration of colorectal cancer cells [50]. *IVL* (involucrin) was the most downregulated gene in the 3D model and was found to be downregulated in most of the tumor samples. Involucrin has earlier been suggested as a promising marker of tumor differentiation and survival in squamous carcinoma of the larynx [51]. In this study, we found a trend towards a better OS for patients expressing high IVL. High expression of α-SMA has been proposed as a marker for activated fibroblasts, and a higher expression of α-SMA has been shown in CAFs compared to normal fibroblasts [52]. However, during the last years, there has been published data showing different populations of CAFs with different properties and with different expression of for example α-SMA and BMP4 [53,54]. In this study, we found a higher expression of *α-SMA* in the HNSCC biopsies than in non-cancerous oral tissue.

The role of CAFs for progression and treatment response of HNSCC is still unclear. The impact of fibroblasts for radiation response largely depends on the combination of fibroblast and tumor cell type, and the same fibroblast type has been shown to either decrease or increase the radiation response in different cancer cell lines [55]. Furthermore, it has been shown that cisplatin response in FaDu cells was CAF-specific when CAFs from seven different HNSCC patients were tested in a 2D model [15]. Our results show that two proteins upregulated of CAFs, *FMOD* and *MMP9*, have a positive impact on OS with regard to radiotherapy; however, due to the small amount of patient material, this has to be further investigated. An increased understanding of the crosstalk between cancer cells and CAFs is of critical importance, as it can lead to the design of new strategies in the treatment of HNSCC. Therefore, we investigated which genes were affected by their patient matched CAFs in a 2D co-culture model and in a more biologically relevant 3D in vitro model. The pattern of CAF localization within the spheroids can have impact on paracrine signaling, which resembles the tumor microenvironment better. Moreover, CAF-mediated remodeling of extracellular matrix in our 3D model manifests with higher expression of collagen and MMPs as compared to 2D model. Interestingly, two of the genes, *MMP9* and *FMOD*, that both were upregulated in tumor cell/CAF spheroids compared to tumor spheroids were found to have a positive impact on overall survival in HNSCC patients. In our recent study, we showed that CAFs increased proliferation and sensitivity to cisplatin in two of three HNSCC cell lines in 3D co-cultures [16]. One explanation could be due on the rationale that the rapidly proliferating cancer cells are more sensitive to DNA-damaging treatment such as ionizing radiation and cisplatin treatment than cells with lower proliferation rate [56].

## 5. Conclusions

In summary, by co-culture of HNSCC cell lines with their patient-matched CAF, the expression of multiple genes was found to be regulated by CAF-derived signals. Moreover, the most changed genes were also found to be differentially expressed in tumor tissue compared to normal oral tissue.

## Figures and Tables

**Figure 1 cancers-13-03361-f001:**
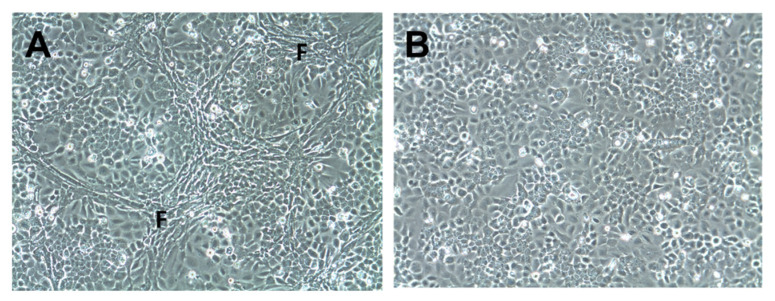
Co-culture of tumor cells and cancer-associated fibroblasts in 2D. Cells of the head and neck squamous cell carcinoma cell line LK0923 were co-cultured with their patient-matched cancer-associated fibroblasts (CAFs) (**A**) or cultured alone; (**B**) for 7 days. Fibroblast-rich areas (F).

**Figure 2 cancers-13-03361-f002:**
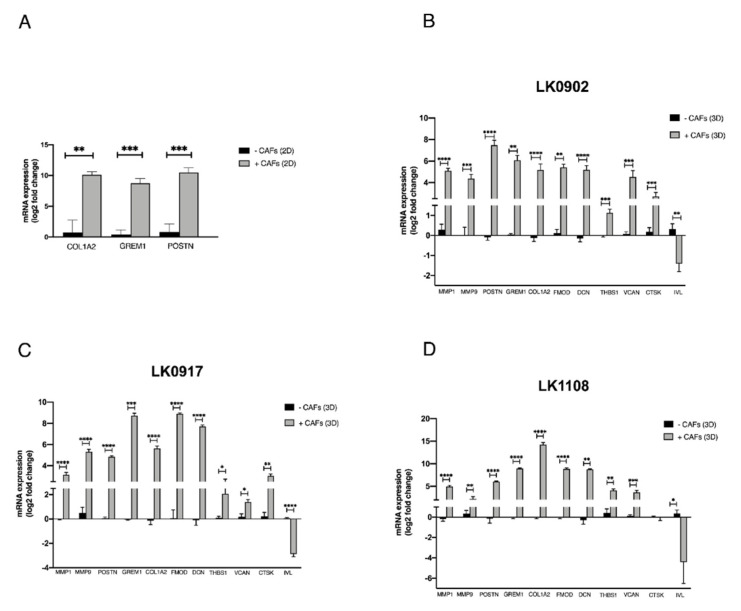
Verification of microarray with qRT-PCR. Five head and neck squamous cell carcinoma cell lines (LK0824, LK0858, LK0923, LK0942, LK0949) were grown alone or co-cultured with their patient-matched cancer-associated fibroblasts (CAFs) as 2D monolayers for 7 days. RNA was isolated from tumor cells (separated from CAFs by magnetic activated cell sorting) and the mRNA expression of the *COL1A2, GREM1*, and *POSTN* genes, which, by microarray analysis, were identified as differentially expressed in co-cultures, was analyzed by qRT-PCR. The log_2_ mRNA levels of co-cultured tumor cells are shown relative to the expression in tumor cells grown alone (**A**). Three head and neck squamous cell carcinoma cell lines were grown in spheroids alone or co-cultured with their patient-matched cancer-associated fibroblasts (CAFs) for 5 days. RNA was isolated from tumor cells (separated from CAFs by magnetic activated cell sorting) and the mRNA expression of the *MMP1, MMP9, POSTN, GREM1, COL1A2, FMOD, DCN, THBS1, VCAN, CTSK*, and *IVL* genes, which, by microarray analysis, were identified as differentially expressed in co-cultures, was analyzed by qRT-PCR. The log_2_ mRNA levels of co-cultured tumor cells are shown relative to the expression in tumor cells grown alone (**B**–**D**). Data were normalized to cells cultured without CAFs for each gene (mean values ± SD; *n* = 3). * *p* ≤ 0.05, ** *p* ≤ 0.01, *** *p* ≤ 0.001, **** *p* ≤ 0.0001 according to Student’s *t*-test.

**Figure 3 cancers-13-03361-f003:**
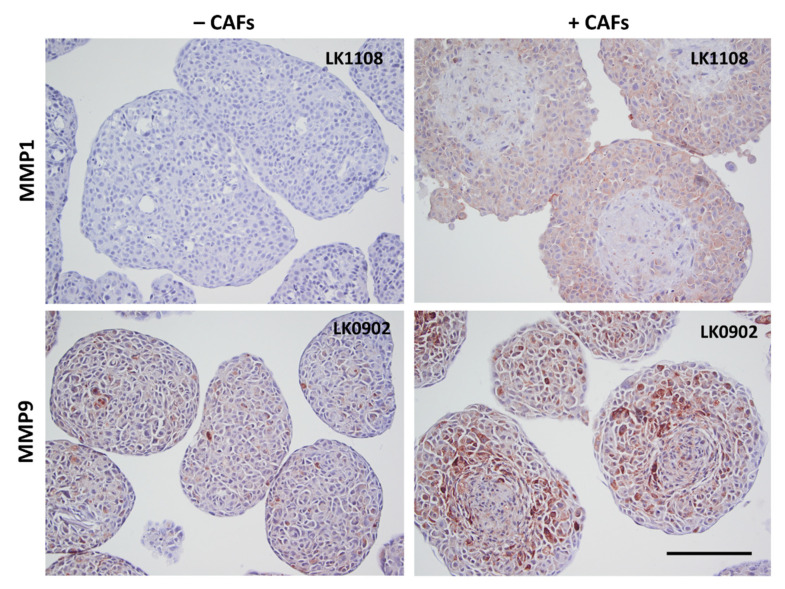
Verification of microarray with immunohistochemistry. Expression of MMP1 and MMP9 in 5-day-old tumor cell spheroids (±CAFs) by immunohistochemical staining. Scale bar = 200 µm.

**Figure 4 cancers-13-03361-f004:**
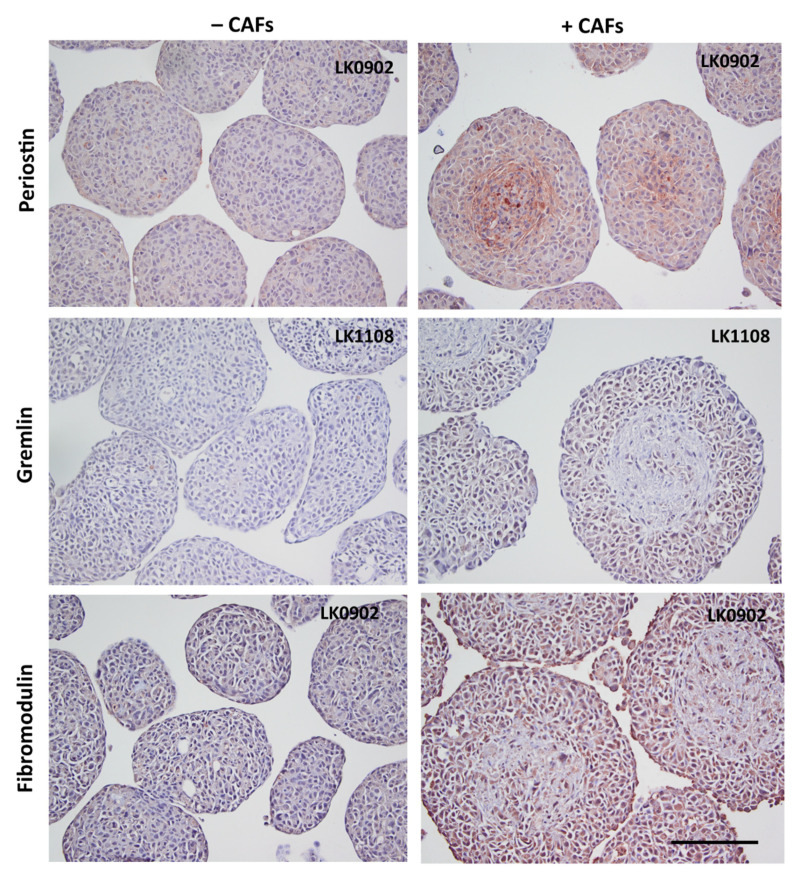
Verification of microarray with immunohistochemistry. Expression of periostin, gremlin, and fibromodulin in 5-day-old tumor cell spheroids (±CAFs) by immunohistochemical staining. Scale bar = 200 µm.

**Figure 5 cancers-13-03361-f005:**
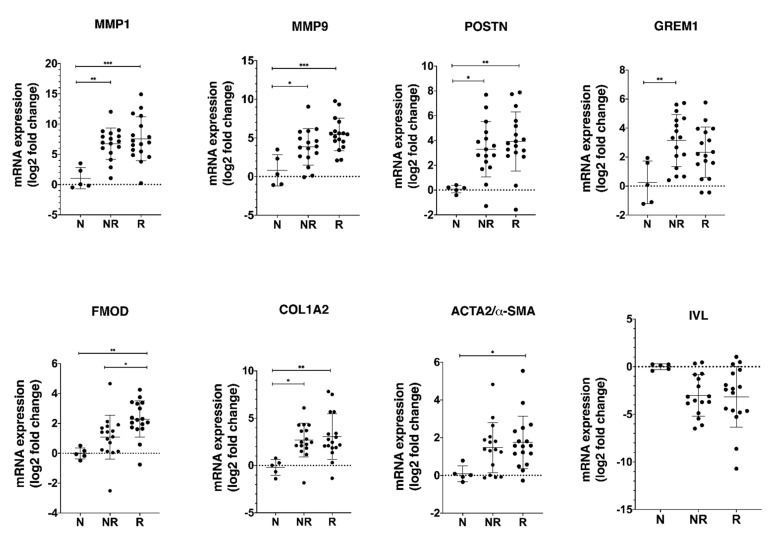
Validation of selected CAF-regulated mRNA candidates in biopsies obtained from patients diagnosed with HNSCC. RNA was isolated from tumor tissue (17 responders, 16 non-responders) and normal oral tissue (*n* = 5) and the mRNA expression of the *MMP1, MMP9, POSTN, GREM1, COL1A2, FMOD*, and *IVL* genes, which, by microarray analysis, were identified as differentially expressed in co-cultures, was analyzed by qRT-PCR. The CAF marker such as α-SMA was also included in the analysis. The log_2_ mRNA levels of tumor tissue are shown relative to the expression in normal oral tissue. Data are shown as mean values ± SD; *n* = 3. * *p* ≤ 0.05, ** *p* ≤ 0.01, *** *p* ≤ 0.001. Abbreviations: N, adjacent normal oral tissue; NR, non-responder; R, responder.

**Figure 6 cancers-13-03361-f006:**
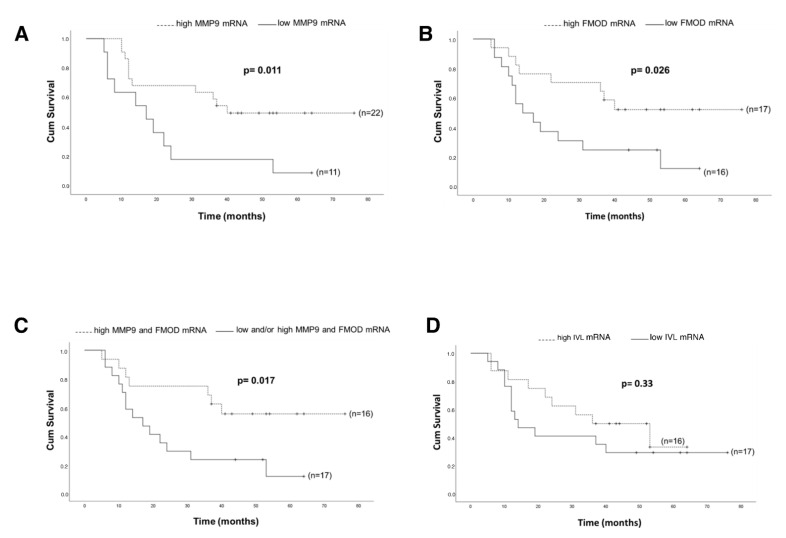
Kaplan–Meier log-rank survival analysis of HNSCC patients with regard to different mRNA expressions. (**A**) *MMP9*; (**B**) *FMOD*; (**C**) *MMP9/FMOD*; (**D**) *IVL*.

**Table 1 cancers-13-03361-t001:** Altered genes in 2D and 3D co-culture of HNSCC cells with CAFs. Only the top 20 genes listed from the 3D experiment.

Gene ID	Log_2_ Fold Change 2D	Gene ID	Log_2_ Fold Change 3D	Gene ID	Log_2_ Fold Change 2D	Gene ID	Log_2_ Fold Change 3D
POSTN	↑ 12.5	MMP1	↑ 152.95	GBP4	↓ −3.12	IVL	↓ −20.64
GREM1	↑ 4.07	MMP9	↑ 49.34	KLK6	↓ −2.7	CNFN	↓ −18.34
COL1A1	↑ 2.46	BGN	↑ 41.26	KLK7	↓ −2.65	KRT1	↓ −16.61
COL1A2	↑ 2.39	COL6A3	↑ 28	SPRR2A	↓ −2.6	DSG1	↓ −13.48
COL6A3	↑ 2.17	POSTN	↑ 27.84	FYB	↓ −2.55	KRT10	↓ −12.64
BGN	↑ 2.08	TNC	↑ 17.97	GABRP	↓ −2.54	MUC15	↓ −10.81
OR4N2	↑ 2	COL1A2	↑ 12.94	ADGRF4	↓ −2.51	SPINK5	↓ −9.66
ID1	↑ 2	IGFBP5	↑ 12.65	IDO1	↓ −2.45	RPTN	↓ −9.35
		FMOD	↑ 11.36	CLDN4	↓ −2.44	CLCA4	↓ −8.91
		PDGFRB	↑ 10.98	GBP6	↓ −2.44	SDR9C7	↓ −8.53
		COL1A1	↑ 10.91	KRT15	↓ −2.36	SPINK7	↓ −8.1
		DCN	↑ 10.18	LIPH	↓ −2.35	LIPK	↓ −7.95
		HAS2	↑ 9.07	CLDN8	↓ −2.34	FOXN1	↓ −7.53
		EDNRA	↑ 8.77	MMP7	↓ −2.33	BNIPL	↓ −6.93
		THBS1	↑ 7.51	SYTL2	↓ −2.32	SCEL	↓ −6.66
		GREM1	↑ 6.67	CRABP2	↓ −2.31	SPRR1A	↓ −6.52
		AMTN	↑ 6.39	FRK	↓ −2.3	CES1	↓ −5.43
		SPARC	↑ 6.19	TMPRSS4	↓ −2.27	MIR203A	↓ −5.06
		EMP3	↑ 5.51	CYP4F3	↓ −2.19	OCLN	↓ −4.45
		CTSK	↑ 3.84	TLR6	↓ −2.16	KLK10	↓ −4.15

↑ upregulated, ↓ downregulated.

## Data Availability

Data generated or analyzed during this study are included in this article.

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
