# Peer review of "Cancer-Associated Fibroblasts Modulate Transcriptional Signatures Involved in Proliferation, Differentiation and Metastasis in Head and Neck Squamous Cell Carcinoma"

_cancers, 2021, doi:10.3390/cancers13133361_

Round 1
Reviewer 1 Report
Cancer-associated fibroblasts (CAFs) stimulate tumorigenesis. Wiechec, Roberg et al. found the effect of CAFs on the gene expression in head and neck squamous cell carcinoma (HNSCC) cells, prepared using 2D and 3D in vitro model systems. CAFs increased the expression of the genes, including MMP9 and FMOD, significantly associated with the overall survival. The paper is well written, and their findings provide the critical information to elucidate how CAFs stimulate tumorigenesis.
Major points:
- Line 323. ‘The expression of α-SMA was higher’. I wish the authors discuss why α-SMA (a CAF marker) expression was higher in the HNSCC biopsies than in adjacent normal oral tissue. The HNSCC biopsies the authors used contain CAFs?
- Lines 421-424. ‘In this study, patients in the responder group have a significant higher expression of FMOD….a low expression (Figure 6B).’ Can the authors connect these findings and discuss the mechanism.
Minor points:
- Line 23. ‘theexpression’ must be ‘the expression’.
- Line 39. ‘Six of these genes’ Can the authors clarify these six are among the thirteen found in 2D, the 81 in 3D, or the genes found in both cases.
- Line 40. ‘MMP9 and FMOD’ Can the authors briefly introduce what they are in the abstract.
- Line 71. ‘cetuximab treatment’ Can the authors briefly explain what that is.
- Table 1. 2nd line ‘1.,5’ must be ‘1.5’.; 6th line ‘2,17’ must be ‘2.17’.
- Line 320. ‘Our result show’ may be ‘Our results show’.
- Line 370. ‘FN1was’ may be ‘FN1 was’.
Author Response
Major points:
- Line 323. ‘The expression of α-SMA was higher’. I wish the authors discuss why α-SMA (a CAF marker) expression was higher in the HNSCC biopsies than in adjacent normal oral tissue. The HNSCC biopsies the authors used contain CAFs?
We have added information and a discussion about α-SMA and CAFs in the “Discussion” section, line 476-489.
- Lines 421-424. ‘In this study, patients in the responder group have a significant higher expression of FMOD….a low expression (Figure 6B).’ Can the authors connect these findings and discuss the mechanism.
We have extended discussion regarding the obtained data about FMOD in the “Discussion” section, line 460-467.
Minor points:
- Line 23. ‘theexpression’ must be ‘the expression’.
This has now been corrected.
- Line 39. ‘Six of these genes’ Can the authors clarify these six are among the thirteen found in 2D, the 81 in 3D, or the genes found in both cases.
We have clarified which genes were upregulated both in 2D and 3D. Line 39.
- Line 40. ‘MMP9 and FMOD’ Can the authors briefly introduce what they are in the abstract.
We have explained in the abstract why MMP9 and FMOD was included. Line 40-41.
- Line 71. ‘cetuximab treatment’ Can the authors briefly explain what that is.
Cetuximab is a monoclonal antibody against EGFR which has now been explained in the Introduction section. Line 78.
- Table 1. 2nd line ‘1.,5’ must be ‘1.5’.; 6th line ‘2,17’ must be ‘2.17’.
This has now been corrected.
- Line 320. ‘Our result show’ may be ‘Our results show’.
This has now been corrected.
- Line 370. ‘FN1was’ may be ‘FN1 was’
This has now been corrected.
Reviewer 2 Report
The authors investigated the influence of cancer-associated fibroblasts over the expression of proliferation, differentiation, and metastasis HNSCC tumor cell genes, compared to the adjacent non-tumoral tissue. The study it is impressive through the amount of data, scientific impact and the experimental design.
Author Response
Dear reviewer,
Thank you for your positive comments to our manuscript.
Reviewer 3 Report
Manuscript describes gene expression differences in HNSCC in the presence or absence of CAFs, both in 2D and in 3D cultures. Some of the deregulated genes were validated using immunohistochemistry in 3D cultures. In addition, expected differences in gene expression were observed between normal oral tissue and tumor samples. Finally, correlation of some of them with clinical outcome is presented from mRNA analysis on clinical samples.
Manuscript results, although confirm that CAFs can change expresion of HNSCC cells, are descriptive and no mechanistic insights onto CAFs-HNSCC are provided.
- Both introduction and discussion are poor, as few previous findings already reported onto CAFs and HNSCC are mentioned. Important results are not discussed.
- Gene expression microarray bioinformatic analyses are not properly described. Corrected p-val (FDR) threshold described in M&M section (FDR>0.1) was not readly used. Instead, no threshold was applied on FDR, but on (uncorrected) p-val. Please clarify why no filtering on FDR was used.
- How many replicates were analyzed per cell line/condition? A proper description of samples included, replicates (biological and/or tehcnical), etc is missing. Authors should show a heatmap of significantly deregulated genes in all samples mentioned in the manuscript. Data should be uploaded to public repositories (GEO).
- Which statistic test was used to extract differences in gene expresion? Ttest? Limma? SAM? Paired Ttest?
- Authors claimed over the manuscript that CAFs contribute to tumor progression, tumor growth, invasive potential, and metastasis. However, the genes described by the authors as overexpressed in HNSCC by the presence of CAFs do not display higher expression in non-responder versus responder patients. Instead, the tendency is the reverse for most of them, and significant in the case of FMOD. These results suggest that gene expression program induced in HNSCC by CAFs might correlate with better outcomes, as shown also in survival curves for MMP9 and FMOD. Therefore, are CAFs contributing to a better or worse disease in HNSCC? I would like the authors to comment on this. Could it be that the role of CAFs is still not fully understood in HNSCC?
Author Response
Manuscript describes gene expression differences in HNSCC in the presence or absence of CAFs, both in 2D and in 3D cultures. Some of the deregulated genes were validated using immunohistochemistry in 3D cultures. In addition, expected differences in gene expression were observed between normal oral tissue and tumor samples. Finally, correlation of some of them with clinical outcome is presented from mRNA analysis on clinical samples.
Manuscript results, although confirm that CAFs can change expresion of HNSCC cells, are descriptive and no mechanistic insights onto CAFs-HNSCC are provided.
- Both introduction and discussion are poor, as few previous findings already reported onto CAFs and HNSCC are mentioned. Important results are not discussed.
-We have extended the “Introduction” with recently published information, line 70-76.
-We have extended the “Discussion” section with discussion about the most important results.
-The results of MMP9: line 442-447.
-The results of FMOD: line 460-467.
-We have added information and a discussion about α-SMA and a discussion about the impact of CAFs in the “Discussion” section, line 476-489 and 500-504.
- Gene expression microarray bioinformatic analyses are not properly described. Corrected p-val (FDR) threshold described in M&M section (FDR>0.1) was not readily used. Instead, no threshold was applied on FDR, but on (uncorrected) p-val. Please clarify why no filtering on FDR was used.
We thank the reviewer pointing this important mistake. Calling of differentially expressed genes was done as described in materials and methods (this section has been re-written) and was based on FDR-corrected p £0.1 and/or fold change < −2 or >2 throughout the manuscript. These rather liberal thresholds were chosen as multiple independently derived HNSCC cell lines were treated as biological replicates in the design, causing higher biological variability than typically seen in such experiments. The manuscript has now been revised in multiple points and we have tried to clarify these aspects.
- How many replicates were analyzed per cell line/condition? A proper description of samples included, replicates (biological and/or tehcnical), etc is missing. Authors should show a heatmap of significantly deregulated genes in all samples mentioned in the manuscript. Data should be uploaded to public repositories (GEO).
We thank the reviewer raising these important questions. Briefly, 2D microarray experiment included in total 5 biological replicates. The 3D experiment in turn included 3 biological replicates. In both cases multiple independently derived HNSCC cell lines were treated as biological replicates. While this design indeed has its pitfalls and may result in type II errors (i.e. false negatives), it allows finding of more robust and general biomarkers and should more likely capture general features that are recurrently reproduced in future research.
The proper description of samples used in 2D and 3D microarray experiments has now been added to the supplementary material, Table S3 and Table S4.
The heatmap of significantly deregulated genes 2D and 3D microarray experiments has now been added to the supplementary material, Figure S7 along with the PCA plot for both microarray designs, Figure S6. We also would like to point out that these figures show that tumor cell lines with and without CAFs form their own clusters that are separated from each other.
Microarray data from 2D and 3D microarray experiments has now been submitted to GEO and the accession numbers is GSE178153 and GSE178154.
- Which statistic test was used to extract differences in gene expresion? Ttest? Limma? SAM? Paired Ttest?
We thank the reviewer pointing out this mistake and not mentioning, which statistical test we used. Differential expression analysis was carried out using Limma with a paired design as implemented in Transcription Analysis Console version 4.0.2.15 (Thermo Fisher). This information has now been added in the Material and methods section
- Authors claimed over the manuscript that CAFs contribute to tumor progression, tumor growth, invasive potential, and metastasis. However, the genes described by the authors as overexpressed in HNSCC by the presence of CAFs do not display higher expression in non-responder versus responder patients. Instead, the tendency is the reverse for most of them, and significant in the case of FMOD. These results suggest that gene expression program induced in HNSCC by CAFs might correlate with better outcomes, as shown also in survival curves for MMP9 and FMOD. Therefore, are CAFs contributing to a better or worse disease in HNSCC? I would like the authors to comment on this. Could it be that the role of CAFs is still not fully understood in HNSCC?
The role of CAFs for cancer progression and treatment response in HNSCC is still unclear. We have now tried to discuss our results according to the reviewer’s suggestion: MMP9: line 442-447 line 460-467 FMOD, line 470-478, CAFs line 489-493. Several recent publications show differences between CAFs from different patients as well as different subgroups of CAFs possessing different properties in regard to treatment response, immune response, migration etc. We have seen in our 3D model that tumor cells growing with CAFs from other patients change the tumor cells response to cisplatin compared with patient matched CAFs (unpublished data). Therefore, we assume that the role of CAFs is not fully understood and more investigations are needed.
Round 2
Reviewer 3 Report
Manuscript has been improved. Introduction, M&M, Results and Discussion are now better.
For publication, minor issues need to be fixed:
- Microarray analysis: in the M&M section, authors say that "...Genes with FDR-corrected p ≤0.1 and/or fold change < −2 or >2 were considered differentially expressed...", which is fine. However, in the results section, the criteria to select differentially expressed genes was only based on fold change. Therefore, sentence in M&M should be modified to "...Genes with fold change < −2 or >2 were considered differentially expressed...". The inclusion of technical replicates of each cell line (at least 3) would have allowed to obtain more significantly deregulated genes (p-val>0.05, FDR<0.1, FC>[2]). As some of the differentially expressed genes (not significant) were validated by other techniques, microarray data can be shown. Otherwise, the defective experimental design and almost absence of significantly deregulated RNAs would have been unaceptable.
- Figure S7: heatmap pattern does not allow visualization of relative differences between groups. Therefore, a z-score transformation of expression values per gene should be performed. Please, for the genes mentioned in the Table 1 and results, indicate their names in their positions in the heatmaps.
Author Response
Manuscript has been improved. Introduction, M&M, Results and Discussion are now better. For publication, minor issues need to be fixed:
- Microarray analysis: in the M&M section, authors say that "...Genes with FDR-corrected p ≤0.1 and/or fold change < −2 or >2 were considered differentially expressed...", which is fine. However, in the results section, the criteria to select differentially expressed genes was only based on fold change. Therefore, sentence in M&M should be modified to "...Genes with fold change < −2 or >2 were considered differentially expressed...". The inclusion of technical replicates of each cell line (at least 3) would have allowed to obtain more significantly deregulated genes (p-val>0.05, FDR<0.1, FC>[2]). As some of the differentially expressed genes (not significant) were validated by other techniques, microarray data can be shown. Otherwise, the defective experimental design and almost absence of significantly deregulated RNAs would have been unacceptable.
We absolutely agree with the reviewer that not having biological replicates weakens the result of microarray analysis. We would like to point that we work with a very limited patient material such as patient derived CAFs. In order to overcome this limitation, using multiple HNSCC cell lines that are cultured in the same conditions (+CAF and -CAF) could give a good picture of CAF-induced changes in the 3D model of HNSCC. Our microarray results have been validated by qPCR, which makes the results reliable.
The sentence in the M&M section has been changed according to reviewer’s suggestion and the word ²significant² in the context of differentially regulated genes has been removed.
- Figure S7: heatmap pattern does not allow visualization of relative differences between groups. Therefore, a z-score transformation of expression values per gene should be performed. Please, for the genes mentioned in the Table 1 and results, indicate their names in their positions in the heatmaps.
We have now prepared heatmaps of the microarray results taking into account the z-score values for each gene in all analyzed samples. Additionally, the names of the most differentially regulated genes have been annotated in the figure S7.